# Green Production of Biomass-Derived Carbon Materials for High-Performance Lithium–Sulfur Batteries

**DOI:** 10.3390/nano13111768

**Published:** 2023-05-30

**Authors:** Chao Ma, Mengmeng Zhang, Yi Ding, Yan Xue, Hongju Wang, Pengfei Li, Dapeng Wu

**Affiliations:** 1College of Mechanical and Electrical Engineering, School of 3D Printing, Xinxiang University, Xinxiang 453003, China; 2School of Business, Henan Normal University, Xinxiang 453007, Chinalipengfei@htu.edu.cn (P.L.); 3Key Laboratory for Yellow River and Huai River Water Environmental and Pollution Control, School of Environment, Ministry of Education, Collaborative Innovation Center of Henan Province for Green Manufacturing of Fine Chemicals, Henan Normal University, Xinxiang 453007, China

**Keywords:** carbon materials, biomass, working mechanism, lithium–sulfur batteries, cathodes, interlayer

## Abstract

Lithium–sulfur batteries (LSBs) with a high energy density have been regarded as a promising energy storage device to harness unstable but clean energy from wind, tide, solar cells, and so on. However, LSBs still suffer from the disadvantages of the notorious shuttle effect of polysulfides and low sulfur utilization, which greatly hider their final commercialization. Biomasses represent green, abundant and renewable resources for the production of carbon materials to address the aforementioned issues by taking advantages of their intrinsic hierarchical porous structures and heteroatom-doping sites, which could attribute to the strong physical and chemical adsorptions as well as excellent catalytic performances of LSBs. Therefore, many efforts have been devoted to improving the performances of biomass-derived carbons from the aspects of exploring new biomass resources, optimizing the pyrolysis method, developing effective modification strategies, or achieving further understanding about their working principles in LSBs. This review firstly introduces the structures and working principles of LSBs and then summarizes recent developments in research on carbon materials employed in LSBs. Particularly, this review focuses on recent progresses in the design, preparation and application of biomass-derived carbons as host or interlayer materials in LSBs. Moreover, outlooks on the future research of LSBs based on biomass-derived carbons are discussed.

## 1. Introduction

Nowadays, human society is facing more and more critical social problems as it is confronted with the ever-growing energy demands and serious environmental crises. In order to efficiently store clean and renewable energy, such as solar, wind, tide, geothermal and other energy sources for sustainable development, research studies on advanced energy storage systems have attracted intense attention worldwide [1,2,3,4,5,6]. Among them, secondary batteries with a high energy density represent a cutting-edge energy storage technology. Traditional lithium batteries, which adopt graphite as the anode material and lithium metal oxide (LiCoO_2_, LiNi_x_Co_y_Mn_1−x−y_O_2_) or lithium phosphate (LiFePO_4_) as the cathode material, demonstrate a theoretical energy density of 400 Wh kg^−1^, which has been widely used in portable electronic devices. After years of research and development, the electrochemical properties of the electrodes of lithium-ion batteries have been reaching their theoretical values, but this still cannot fully meet the needs for energy storage devices to power electric vehicles or to store the huge volume of electricity generated from clean-energy harnessing facilities. 

Therefore, researchers have turned their attention to other cathode materials with a high theoretical capacity. Among various potential cathode materials, lithium–sulfur batteries (LSBs) have attracted much attention as a potential low-cost and efficient energy storage system due to the advantages of high theoretical capacity (1675 mAhg^−1^), high energy density (2600 Whkg^−1^), wide sources and low cost of elemental sulfur [7,8]. LSBs consist of elemental sulfur as the cathode, an electrolyte, a separator and lithium metal as the anode. Through the multi-electron electrochemical conversion between sulfur and lithium, LSBs could give rise to a much higher specific capacity than that of traditional lithium batteries.

Carbon-based materials represent a group of important raw materials for different industries. Duo to their high electroconductivity and high thermal and chemical stability, as well as bio-compatibility, many types of carbon materials, such as graphene or reduced graphene oxides [4,5,6,7,8,9,10,11,12,13,14,15], carbon fibers [16,17,18], carbon dots [19], active carbons [20,21,22] and biomass-derived carbons [23,24,25,26,27,28,29,30,31,32], have been deliberately prepared for different application aims. Biomass has many definitions, such as biodegradable products, wastes and residues from agriculture, forestry and related industries, including fisheries and aquaculture, as well as biodegradable parts of industrial and municipal wastes. In addition, biomass is also regarded as a class of organic macromolecular materials derived from organisms [33]. During carbon preparation, biomass precursors undergo thermochemical transformation under high-temperature conditions, which could be classified into two distinctive processes. The first one is solid-state carbonization, which represents the decomposition of biomass precursors under a high temperature and an inert atmosphere to generate carbon-rich and thermal stable products [34]. The second one is hydrothermal carbonization, which could convert wet biomass into a carbon-rich product (hydrochar). In these processes, the organic components in the biomass experience drastic chemical changes, such as carbon skeleton recombination and functional group decomposition, which finally yield a carbon atom network with high-conductivity sp^2^ domains and rich surface functional groups [34]. Generally, activation processes are introduced to further treat the as-obtained carbon materials to increase the surface area and to regulate the pore structures. These activation processes are usually carried out under high-temperature conditions and with the assistance of physical (water vapor or carbon dioxide) or chemical activators (KOH, H_3_PO_4_, ZnCl_2_, NaOH, etc.) [35,36]. In addition, the templating method is also one of the commonly used strategies to optimize the surface area and pore structures of as-obtained carbon materials. Generally, templates are firstly introduced into a biomass precursor, which is then carbonized under a high temperature with inert atmosphere. Finally, the templates are removed by immersion in a NaOH or HF solution to generate carbon materials with ordered pore distribution [37].

Activated carbon production from biomass has long been regarded as important research in materials science. At the very beginning, biomass-activated carbon was only used for adsorption. Thanks to their rich pore structures, huge specific surface area, and good physical and chemical stability, biomass-derived carbon materials have gradually developed into a wide range of adsorbents for application in purification, deodorization, decolorization and separation [38]. China is a traditional agricultural country, and there are abundant biomasses produced from agriculture, forestry, animal husbandry and aquaculture annually. In addition, illegally discarded or wrongly treated biomass has become an important source that causes environmental pollution. Therefore, the production of functional carbon materials from biomass could not only lead to economic benefits but also alleviate such environmental problems, which has attracted more and more attention.

At present, biomass-derived carbon materials are employed in energy storage applications, such as supercapacitors [39], lithium-ion batteries [40,41], lithium–sulfur batteries (LSBs) [42,43], and so on. As for the application of biomass-derived carbons in LSBs, many review works have been devoted to highlight the significance of these materials from different aspects [44,45,46,47,48,49,50,51,52]. This review work will focus on discussing recent advancements in the design, preparation and application of biomass-derived carbons as host or interlayer materials in LSBs. Moreover, outlooks on rational preparation based on biomass-derived carbons and potential future research on LSBs are also discussed.

## 2. Fundamentals

### 2.1. Working Principles of LSBs

In the discharge curve shown below, two typical discharge platforms at 2.3 and 2.1 V could be clearly observed, corresponding to the solid (S_8_), liquid (Li_2_S_n_) and solid (Li_2_S_2_/Li_2_S) processes, respectively [53] (Figure 1a). S_8_ is firstly converted into long-chain Li_2_S_n_, and then the long-chain Li_2_S_n_ is converted to Li_2_S_2_/Li_2_S. For the higher-voltage discharge platform located at 2.3 V, S_8_ is reduced to long-chain polysulfide, corresponding to the theoretical capacity of 418 mAh g^−1^. For the lower-voltage discharge platform at 2.1 V, polysulfide is further reduced to insoluble Li_2_S_2_/Li_2_S, corresponding to the theoretical capacity of 1254 mAh g^−1^. During the charging process, 2.3–2.4 V could be regarded as a platform, corresponding to the transformation process from Li_2_S_2_/Li_2_S to S_8_. The intermediate polysulfide dissolves and diffuses into the electrolyte due to the concentration gradients and electric field forces (Figure 1b). In addition, a small polarization voltage can be observed during the initial phase of charging, which is due to the final product of the discharge phase (Li_2_S_2_/Li_2_S) being an insulator, thus leading to a large inter-phase transition barrier.

### 2.2. Obstacles for LSBs

Due to their high theoretical capacity and energy density, LSBs have attracted extensive attention from both domestic and overseas researchers. However, LSBs still suffer from various inherent technical bottlenecks as listed below, which greatly restricts their application. The first bottleneck is the insulation of the charge and discharge products. Under room temperature, elemental sulfur (S_8_) is an insulator for Li^+^ and electrons (5 × 10^−28^ S cm^−1^). Likewise, Li_2_S is also an insulator for both Li^+^ and electrons (<10^−14^ S cm^−1^) [54]. In the working process, the low Li^+^ and electron conductivity of the charge–discharge product hinders the charge–discharge process of the battery. When sulfur is used alone as the electrode material, such property leads to two serious problems which restrict the development of lithium–sulfur batteries: the low utilization rate of active substances and the slow reaction kinetics. The intrinsic insulation of S_8_/Li_2_S severely limits the utilization rate of S active substances, resulting in low capacity and low-rate capacity. Although the solid–liquid reaction with a discharge platform of about 2.3 V has a faster reaction kinetics, it is still difficult to achieve the theoretical capacity. Additionally, previous reports have demonstrated that after multiple cycles, there is still S_8_ that is not involved in the reaction. In addition, due to the limited Li^+^ and electro-kinetics, in the charge–discharge curve of lithium–sulfur batteries, a large polarization phenomenon is often observed on the lower discharge platform, which further leads to the attenuation of the actual energy density.

The second bottleneck is the shuttle effect of intermediate products. Due to the transport of Li^+^ between the two electrodes, the electrode reaction of lithium–sulfur batteries involves a series of reversible lithiation and delithiation processes. In this series of transformation, the production of soluble Li_2_S_n_ (n ≥ 4) is inevitable, which leads to a series of negative effects [55]. Due to the concentration gradient, the high-valence polysulfide Li_2_S_n_ (n ≥ 4) will migrate to the anode, react with lithium metal at the anode, and then generate Li_2_S_2_/Li_2_S. Therefore, during the charge–discharge process, polysulfide cannot completely migrate back to the cathode, which results in an irreversible loss of active materials. Afterward, the continuous deposition of Li_2_S_2_/Li_2_S on the surface of the lithium negative electrode leads to the continuous generation of solid electrolyte interface film (SEI film) on the anode surface, which hinders the release of Li^+^ and the effective contact between the electrolyte and the negative electrode, resulting in a continuous increase in battery impedance with an increase in the number of cycles. As the dissolution and deposition of active substances redistribute after longtime recycles, a passivation layer forms on the surface of the positive electrode material, which greatly increases the surface resistance of the electrode. The production of soluble Li_2_S_n_ (n ≥ 4) increases the viscosity of the electrolyte and reduces the conduction rate of Li^+^ and electrons in the electrolyte, which poses negative impact on the rate capability of the battery. Therefore, due to the severe shuttle effect, a battery can only maintain a limited number of cycles and experiences poor cycle stability, low coulomb efficiency and self-discharge. Because the density difference between S_8_ and Li_2_S is great, active species of the cathode exhibits a volume expansion of nearly 80% during the discharge process. Such a massive volume expansion leads to two invisible problems. The first one is that the severe volume expansion triggers pulverization and shedding of active materials from the current collector. Meanwhile, the volume expansion also causes a series of safety problems, which seriously limits the practical application of LSBs.

## 3. Recent Progresses in Carbon Materials for LSBs

As we have stated before, the cathode material of LSBs plays an important role in these storage devices. In order to solve the aforementioned technical bottleneck and realize the commercial production of lithium–sulfur batteries, a large number of scholars have dedicated efforts to optimizing the LSB system from the perspective of cathode materials, hoping to make up for the intrinsic defects of lithium–sulfur batteries based on the design of cathode materials. According to recent research on carbonous cathodes for LSBs, we summarize recent progresses into two categories: structural regulation and heteroatom doping of carbonous materials.

### 3.1. Structural Regulation of Carbons

Carbon materials exist widely in the natural environment and have stable physical and chemical properties. The introduction of carbon materials can significantly improve the electronic conductivity and ion transport and buffer the volume expansion of active materials, which could avoid the pulverization and shedding of the positive electrode structure in the process of charge and discharge. Therefore, as shown in Figure 2, the exploration of carbon/sulfur composite cathode materials has triggered great research attention, and carbon materials, including porous carbon, hollow carbon structure, graphene, and so on, are commonly used for the cathode in lithium–sulfur batteries [53].

**Porous carbon.** Porous carbon is a functional carbonaceous material with porous structure. According to the pore size distribution, porous carbons could be classified into microporous carbon materials (pore size less than 2 nm), mesoporous carbon materials (pore size between 2 and 50 nm), macroporous carbon materials (pore size greater than 50 nm), and hierarchical porous carbon materials (with a variety of pore structures). High porosity and high specific surface area are conducive to the storage and uniform distribution of sulfur. It is also proposed that a porous structure could lead to better inhibition of the dissolution and diffusion of polysulfides, which effectively reduces the shuttle effect and, thus, improves the electrochemical performances of LSBs [47,48].

As shown in Figure 3a, sucrose is used as a carbon source to obtain uniform microporous carbon spheres with a specific surface area of 843.5 m^2^ g^−1^ and a pore size distribution of mainly 0.7 nm [56]. When the sulfur content is 42 wt%, the initial discharge capacity could amount to 1183.5 mAh g^−1^, and after a long period of cycling, it still has a reversible capacity of 650 mAh g^−1^. The electrode only shows a discharge platform at 1.8 V, which is not a typical lithium–sulfur battery with two discharge platforms, and the high-voltage discharge platform is absent. The unique phenomena could be interpreted as that due to their small size, sulfur molecules are the main species stored in the micropores of the carbon matrix, which avoids the generation of soluble polysulfide and results in a typical charge–discharge curve. As demonstrated by the theoretical simulation (Figure 3b) [7], when the size of microporous carbon is small enough (0.5 nm), S_8_ molecules could be split into small sulfur molecules with a short chain length and stored in the micropores, which could inhibit the transformation from S_8_ to S_4_^2−^, alleviate the shuttle effect of polysulfides, and, finally, result in the high capacity retention and coulombic efficiency during the charge–discharge process. Highly ordered mesoporous structures with a pore size of 3–4 nm are also synthesized as a conductive matrix to accommodate the S loading. During the charge and discharge process, this structure promotes the reaction between Li^+^ and sulfur and inhibits the diffusion of soluble polysulfide by trapping polysulfide within the carbon framework [57].

**Carbons with hierarchical porosities.** Microporous carbon has a high specific surface area, which can ensure the dispersion and contact of elemental sulfur in the conductive matrix. Moreover, the strong physical adsorption capacity of micropores can effectively inhibit the shuttle effect. However, micropores can only provide a limited pore volume, which makes it difficult to accommodate higher active substances. When sulfur loading exceeds the critical amount, the extra sulfur will harm the electrical contact with the conductive matrix, which will reduce the utilization rate of active substances and limit the overall energy density of the battery. Large pores and mesoporous pores can house more sulfur than that of the micropores, which could substantially increase the sulfur loading amount of the electrode and effectively alleviate the volume expansion during the charge and discharge process. However, physical adsorption is not capable enough in inhibiting the dissolution and diffusion of polysulfides, thus resulting in an irreversible loss of active substances. Therefore, the design of hierarchical porous carbon materials with various pore size distributions could well balance the advantages of macroporous, mesoporous and microporous carbon materials and, thus, lead to improved electrochemical performances.

Liang et al. used ordered mesoporous carbon as the precursor and KOH as the activator to obtain hierarchical porous carbon with a dual-pore distribution (3 nm and 7.3 nm) (Figure 4a) [58]. Sulfur is found evenly distributed in the conductive carbon matrix, in which the micropores serve as the storage space of active sulfur to ensure the contact between sulfur and the conductive matrix. Mesoporous pores not only could effectively accommodate polysulfides dissolved in the electrolyte but also provide a fast transport channel for lithium ions. In addition, a layer of microporous carbon is coated on the surface of the highly ordered mesoporous structure to obtain a hierarchical porous carbon material with a core–shell structure (Figure 4b) [59]. The ordered mesoporous could greatly improve the sulfur loading amount and make full utilization of the carbon matrix. Meanwhile, the microporous carbon shell could function as a polysulfide barrier to reduce the capacity decay of the battery. Based on this, a rapid spray drying method was adopted to obtain carbon spheres with a hierarchical porous structure (Figure 4c) [60]. Mesoporous and microporous structures could be introduced inside the carbon spheres, which helps to achieve high sulfur accommodation. The outer microporous shell could be used as a physical site to anchor polysulfides to avoid the irreversible loss of active substances induced by the diffusion of polysulfides.

**Core–shell structure and hollow structure:** Core–shell nanocomposites can be employed as physical prevention to inhibit the diffusion of polysulfide. On the other hand, polysulfide can be effectively bonded to core–shell structures, which could inhibit the shuttle effect of polysulfide. Meanwhile, the hollow structure provides a large inner cavity, and the shell usually acts as a barrier layer, which could greatly enhance the loading amount of active materials and effectively inhibit the dissolution and diffusion of polysulfides. However, the sulfur species trapped in the inner cavity of the hollow structure are not easily accessible. Therefore, many researchers have tried to optimize the internal cavity to realize the anchoring of polysulfides while ensuring the rapid transfer of the electrolyte, electrons and Li^+^ ions toward the active species.

Due to its insulating properties and the shuttle effect of polysulfides, Li_2_S exhibits poor utilization of active substances and a short cycle life. Based on these, core–shell Li_2_S@C nanocomposites were prepared using plasma sparking and subsequent vulcanization process, through which Li_2_S particles could be uniformly coated with a carbon layer of 0.8 nm [61]. It was demonstrated that the carbon coating effectively increases the conductivity of the composite material and effectively reduces the shuttle effect, resulting in superior electrochemical performances (Figure 5a). In addition, a solution evaporation method collaborated with a CVD process was adopted to prepare core–shell Li_2_S@C composites (Figure 5b) [62]. The homogeneous amorphous carbon layer with a thickness of about 20 nm was evenly coated on Li_2_S, which effectively prevents Li_2_S from agglomeration. Compared to the pure Li_2_S electrode, the core–shell Li_2_S@C composite cathode shows high utilization rate of active species and excellent electrochemical performances. In addition, core–shell carbon spheres were prepared via the templating method, which compose of a mesoporous shell, a hollow cavity and a fixed carbon core (Figure 5c) [63]. When used as a carbonaceous sulfur carrier, the hollow cavity can improve the sulfur loading amount and reduce the volume expansion during the charge–discharge process. The mesoporous shell can provide a Li^+^ transport channel and inhibit the shuttle effect of polysulfide. In general, this core–shell structure could function as both a physical buffer and a conductive matrix to maximize the potential capacity of active substances and alleviate the intrinsic deficiencies of LSBs.

**Graphene:** Graphene represents a new kind of two-dimensional material with a hexagonal lattice structure composing of sp^2^ carbon domains. The π-π conjugated bonding among carbon atoms endows graphene with excellent electrical conductivity. In addition, graphene with a high surface area, high conductivity and stable physical and chemical properties could serve as a potential candidate for LSBs. For example, a simple one-step method was developed to anchor sulfur nanocrystals to 3D cross-linked fibrous graphene (Figure 6a) [64]. When used as the cathode material for Li–sulfur batteries, the porous three-dimensional conductive network and uniformly distributed sulfur nanocrystals achieve rapid electron transport and shorten Li^+^ diffusion distance. In addition, the presence of oxygen-containing functional groups enhances the anchoring ability of polysulfide and prevents the dissolution of polysulfide in the electrolyte, which amounts to the high capacity, high-rate performance and long cycle life. Moreover, graphene foam electrodes were proposed and prepared through an effective strategy to obtain flexible Li–sulfur batteries with high energy and power densities as well as long cycle life (Figure 6b) [65]. This research found that graphene foams can provide highly conductive networks, strong mechanical support and enough room for a high sulfur loading amount. In order to further enhance the electrochemical performances, hollow nanographene spheres (GSs) supported by carbon nanotubes (CNTs) were prepared using a room-temperature solubility-processable method (Figure 6c) [66]. Within the unique flexible electrode, the conductive carbon nanotubes could serve as flexible scaffolds and the hollow GSs provide a closed space to accommodate the sulfur species, which could adapt to the volume expansion and inhibit the shuttle and dissolution of polysulfides, leading to rapid electron and ion transport.

### 3.2. Heteroatom-Doped Carbons for Cathode Materials

Nonmetallic heteroatoms (N, O, S, P, etc.) can be used as anchoring sites for polysulfides. Therefore, in addition to structural design, the introduction of heteroatoms into the conductive carbon matrix is also an effective way to improve the performance of batteries. Due to the electronegativity difference, electron-rich heteroatoms can lead to the surface polarization of the non-polar conductive carbon matrix, which helps to improve the chemical anchoring ability of the carbon matrix to polysulfide. At the same time, heteroatom impurity can introduce defect sites to enhance the catalytic performances of the electrodes so as to enhance the conversion of polysulfides. As shown in Figure 7a, nitrogen-doped mesoporous carbon microspheres permeated by carbon nanotubes as sulfur hosts were prepared through a self-assembly method, which greatly improves the performance of the batteries [67]. Due to the interaction between Li^+^ and N atoms, soluble polysulfide in the electrolyte can be anchored by forming stable Li_2_S_X_-N chemical bonds, and polysulfide can be confined to the cathode materials, which significantly inhibits the shuttle effect and improves the electrochemical performance of the battery. Unlike conventional insulating adsorbents, nitrogen doping can inhibit the shuttle effect through the chemical interaction between the doping site and polysulfide. At the same time, the high conductivity of the nitrogen-doped carbon materials can directly trigger the conversion of polysulfides at the electrodes. For example, boron is a typical electron-deficient element, which can interact with electron-rich sulfur and polysulfide to form stable chemical bonds and, thus, can be used to absorb polysulfide and inhibit the shuttle effect. Han et al. prepared a boron-doped porous carbon material as the sulfur carrier [68] (Figure 7b). Compared to the undoped porous carbon, the boron-doped material showed high initial capacity (1300 mA h g^−1^, 0.25 C), good cyclic stability and superior rate performance. Based on the mechanism study, boron, with a lower electronegativity than carbon, provides positively charged active sites, which could effectively adsorb negatively charged polysulfide.

Compared to the single doping effect, the co-doping of heteroatoms can effectively combine the advantages of different heteroatoms, which can greatly improve the conductivity of carbon matrix and the adsorption of polysulfide. Diketoxime (DMG) and nickel chloride tetrahydrate (NiCl_2_·4H_2_O) were used as precursors for the preparation of nitrogen and oxygen co-doped porous carbon microrods with a large specific surface area and high porosity (Figure 8a) [69]. Based on the density functional theory (DFT) calculation, it is confirmed that the introduction of nitrogen and oxygen heteroatoms can effectively improve the adsorption capacity of the carbon matrix toward polysulfides and greatly inhibit the shuttle effect, which significantly improves the electrochemical performance of LSBs. Graphene oxide nanoribbons were modified by adding boric acid and urea to obtain nitrogen and boron co-doped curved graphene nanoribbons [70]. It was found that the reaction of the boric acid/urea precursor could promote the co-doping process of nitrogen and boron. In addition, the rich N-B motifs could significantly improve the electron conductivity, sulfur dispersion and polysulfide adsorption capacity of the electrodes. Wang et al. developed an effective strategy to improve the electrochemical performance of sulfur electrodes via the preparation of nitrogen/sulfur co-doped graphene matrix for the cathode material of LSBs [71]. In addition to the chemical anchoring of polysulfide by the nitrogen and sulfur defect sites, nitrogen and sulfur atoms with high electronegativity lead to the polarization of adjacent carbon atoms and oxygen-containing functional groups, which could increase the adsorption activity of sulfur and polysulfide. At the same time, the highly developed defects and edges and the porous structures obtained via the chemical activation of graphene not only achieve good dispersion of sulfur, but also act as a polysulfide reservoir to mitigate the shuttle effect. In addition, lithium iron phosphate nanoparticles were adopted as a hard template to prepare nitrogen, oxygen and phosphorus co-doped hollow carbon nanocapsules/graphene composites as the sulfur cathode [72]. The shuttle effect of polysulfide could be greatly inhibited by the physical and chemical adsorption of the abundant surface polar groups on the composite material.

## 4. Biomass-Derived Carbon Materials for LSBs

### 4.1. Advantages of Biomass-Derived Carbon Materials

Compared to other non-renewable carbon sources, biomass can be used as an abundant and sustainable carbon source to prepare porous carbon materials for energy storage and conversion [73]. So far as we know, a large variety of biomasses have been used as biomass carbon sources to prepare carbon materials as sulfur carriers or functional separators in lithium–sulfur batteries [74]. Compared to other carbon sources, biomass precursors have the following advantages. First of all, after thousands of years of evolution, biomass usually possesses a unique structure and morphology. The inherent hierarchical channel structure obtained after carbonization is conducive to sulfur accommodation and adsorption, which could potentially alleviate the volume expansion of cathodes. Secondly, biomass precursors have diversified compositions. In the process of carbonization, a biomass’s inherent heteroatoms could be doped into a carbon matrix, which could enhance the conductivity and the adsorption of polysulfides through the strong chemical interactions. Finally, due to the massive production of biomasses annually from different industries, biomass-derived carbons with a low price could substantially reduce the commercial production cost of LSBs.

### 4.2. Biomass-Derived Carbons for the Cathode of LSBs

Biomasses, such as agriculture wastes [75,76,77,78,79], forest wastes [80,81,82,83], weeds [84,85,86], food residues [87,88,89,90,91,92,93,94,95,96,97,98,99,100], and so on, have been developed as precursors to prepare high-performance carbons as cathode hosts for LSBs. As shown in Figure 9a, N, O co-doped carbon with a hierarchical porous structure was derived from bagasse, which could serve as a novel sulfur host for stable LSBs. It was found that the interconnected hierarchical porous structure facilitates the charge transport and alleviates the volume expansion of sulfur during the lithiation process, which finally amounts to highly stable LSBs [75]. In addition, a novel biomass waste, garlic peel, was used as a precursor to prepare carbons through two methods, pre-carbonization and hydrothermal treatment (Figure 9b). Due to the high surface area of carbon, the cathode exhibits high initial specific capacity and cycle retention. These structure advantages could also lead to the intimate contact between sulfur and the conductive carbon matrix, which could physically confine lithium polysulfide intermediates and reduce the shuttle effect [79]. Eucommia leaf residue was employed to prepare carbons with a hierarchical porous structure via the co-auxiliary activation of KCl and CaCl_2_ with a low dosage of KOH (Figure 9c). The optimized pore distribution, high specific surface area and nitrogen-containing functional groups could enhance the utilization of sulfur and provide a chemical anchor for polysulfides, which gives rise to the excellent electrochemical performances [82]. Moreover, carbon materials with rational tailored morphology and structures could be obtained from balsa waste (Figure 9d). It was found that the mesopores in such carbon materials exhibit more merits than micro/macropores in improving sulfur utilization and restraining Li_2_S_x_, which could alleviate the notorious shuttle effect. In addition, the mechanism studies show that the conversion from long-chain polysulfide into solid S_8_ and Li_2_S could be accelerated by oxygen groups, which finally leads to improved sulfur immobilization and stable energy-storage capacity [83].

In addition to these biomasses from agriculture, forest wastes and food residues have also been employed as raw materials for the production of carbon as a host for LSBs. As depicted in Figure 10a, nitrogen/sulfur co-doped porous carbons were manufactured from cattail biomass through a one-step hydrothermal method. The stable foam-like porous structure, high specific surface area and N/S atom-doping sites could greatly inhibit the volume expansion of sulfur and the shuttle effect due to the physical confinement and chemical adsorption during the electrochemical process of LSBs [84]. In addition, nitrogen-doped porous carbon was prepared through the carbonization of pomelo peels to serve as a sulfur host material for LSBs (Figure 10b). The N-doping sites and the hierarchical porous architecture render the carbonous material with excellent sulfur confinement property due to the combination of physical and chemical adsorptions. Therefore, the sulfur composite cathodes exhibit ultrahigh initial capacity, high coulombic efficiency and stable sulfur electrochemistry [88]. Moreover, starch was adopted as a precursor to prepare porous carbon materials without using additional physical and chemical activators [97]. In order to avoid damage to the material structure caused by rapid generation and aggregation of water vapor during pyrolysis, the heating rate and airflow velocity during pyrolysis were carefully controlled (Figure 11a). The as-synthesized carbon material possesses the inherent porous structure of starch, which effectively increases the pore volume of the as-obtained carbon material. It was found that the narrow and long microporous channel not only avoids the direct contact between the electrolyte and active substances, but it also immobilizes polysulfide in the porous shell through physical adsorption, which effectively inhibits the shuttle effect of soluble polysulfide. To further optimize the pore structure of carbon products, mesoscale silica spheres with a uniform size were used as hard templates, which endows the as-obtained carbon with ordered structure and uniform pore size (Figure 11b). The abundant mesoporous structure provides enough space for the storage of active substances and alleviates the volume expansion during lithiation, which greatly overcomes the electronic insulation of sulfur and effectively inhibits the migration of polysulfides [98]. Nitrogen and oxygen co-doped porous carbon materials could also be obtained from soybeans [100]. The protein contained in this precursor is converted into different nitrogen-containing components (pyrrole nitrogen, pyridine nitrogen and graphite nitrogen) during pyrolysis, which leads to a higher nitrogen content in these electron donors to improve the overall electron density of the carbon materials and enhances the conductivity of the carbon materials. In addition, nitrogen-doping sites can be employed as electron-donating sites to bind electron-deficient polysulfides with strong chemical bonds, thus enhancing the anchoring ability of the carbon framework to polysulfides (Figure 11c). Based on the traditional Chinese expansion method, rice was employed as a precursor to prepare porous carbons composed of nickel-doped hybrid nanoflakes [99] (Figure 11d). After expansion, the dense starch structure becomes loose due to steam evaporation. It was found that the final carbon with a sheet-like structure provides a stable three-dimensional porous structure, which avoids the structural collapse caused by volume change during the charge and discharge process. The three-dimensional porous structure, as the space for storing active substances, ensures the electrical contact between the active substances and the conductive network. In addition, the stable chemical bond between Ni/NiO and polysulfide effectively alleviates the shuttle effect and ensures a high-capacity retention rate. Meanwhile, the embedded nickel nanoparticles not only significantly increase the electronic conductivity but also provide a shortened ion diffusion channel to accelerate the diffusion process. In order to further highlight the recent developments in biomass-derived carbons adopted as cathodes in LSBs, a systematic comparison table is provided in terms of the preparation method and the electrochemical performances of LSBs (Appendix A).

### 4.3. Biomass-Derived Carbons for the Interlayer of LSBs

In order to enhance the performance of LSBs, free-standing or deposited interlayers, which could effectively prohibit the shuttle effect of polysulfides, have been designed and introduced between the electrodes without affecting the integrity of LSBs. So far as we know, many types of biomass-derived carbons have been adopted to modify the pristine separator or construct a free-standing interlayer between the electrodes of LSBs [101,102,103,104]. As depicted in Figure 12a, N, O co-doped carbons with a hierarchical porous structure, a high specific surface area, and good electrical conductivity were synthesized from chlorella biomass through a chemical activation process. As they are deposited on the polypropylene separator as an interlayer in LIBs, these carbons can improve the electrolyte wettability and Li^+^ diffusion. In addition, N, O heteroatoms and the porous structure exhibit strong chemical adsorption and provide physical barriers confining lithium polysulfides, which result in enhanced cycling stability and rate performance. As shown in Figure 12b, Ginkgo Folium was employed as a biomass to synthesize carbon materials to decorate the separator of LSBs. Thanks to the three-dimensional interconnected porous structures and the defective graphite structure, the interlayer effectively inhibits the shuttle effect and boosts the electrochemical properties of the LSBs. In addition, rotten egg albumen-derived carbon was also fabricated via freeze drying and carbonization (Figure 12c). As modified on the separator, the layered carbon materials with high conductivity and the rich nitrogen-doped sites could enhance both lithium ion and electrical conductivity, and improve polysulfide adsorption, which substantially inhibits the shuttle effect of polysulfides and enhances the electrochemical performance of LSBs.

As shown in Figure 13a, discarded crab shells were employed as a precursor to yield functional carbon materials through potassium hydroxide-assisted pyrolysis [105]. Due to the intrinsic rich keratin content contained in the crab shells, nitrogen doping could be easily introduced during pyrolysis. As deposited on the membrane surface of the separator, the surface wettability is greatly improved and strong chemisorption of polysulfides is formed, which could effectively enhance the electrolyte permeability, the transportation of lithium ions and the adsorption of the modified separator. As depicted in Figure 13b, bamboo chopsticks were used to prepare carbon fibers as an interlayer for LIBs with excellent electrochemical performances [106]. When used as an interlayer for LSBs, soluble polysulfides can be effectively adsorbed in the bamboo carbon fiber sandwich membrane, and the shuttle effect is substantially suppressed. At the same time, the cross-linked three-dimensional conductive framework constructed by the fiber structure also provides an interworking conductive network, which promotes both the electronic transport and the diffusion rate of lithium ions. In addition, sulfur-doped microporous carbon materials could be prepared from loofah pulp biomass. The as-prepared self-supporting porous carbons could effectively prevent the diffusion of polysulfides to improve the electrochemical performance of LSBs [107]. The mechanism studies indicated that the rich microporous structure and sulfur-doping sites could be used as both physical and chemical adsorption sites, which could effectively adsorb polysulfides in the conductive framework and avoid the irreversible loss of active substances, leading to high sulfur utilization and high cycle stability. Recent research progresses regarding biomass-derived carbons adopted as an interlay in LSBs are summarized in Appendix A.

### 4.4. Modified Biomass-Derived Carbons for LSBs

The main advantages of biomasses rely on their intrinsic heteroatoms and porous structure, which could endow the as-obtained carbonous materials with a high surface area, a hierarchical porous structure and heteroatom-doping sites. However, these merits highly depend on the inherent qualities of the biomass used, which is not favorable for further property enhancement of carbon materials. Therefore, many schemes have been developed to modify the biomass used prior to or after the carbonization process [108]. As shown in Figure 14a, in order to introduce P doping sites on the carbon materials, H_3_PO_4_ was added in cotton stalk biowaste, which could both enhance the surface area and introduce hierarchical pores. The P doping sites in the carbon networks not only provide more active sites but also improve the electrical conductivity, which finally results in excellent electrochemical performances [109]. In addition, CoO nanoparticles were decorated on hierarchical porous carbon by mixing a natural nori and Co precursor prior to the carbonization process. It was found that the carbon substrate could well accommodate CoO nanoparticles, which could enhance the adsorption immobilization of lithium polysulfides and facilitate their redox conversion. Therefore, the composite cathode possesses a high discharge capacity, excellent rate performance and cycling stability [110,111,112,113,114,115,116]. As shown in Figure 14b, rice straw was developed to fabricate conductive biochar and then decorated with highly dispersed CoO nanoparticles via a microwave-assisted method. The high LSB performances are derived from the excellent conductive framework of the biochar and the excellent adsorption capability of CoO nanoparticles, which greatly alleviate the shuttle effect as well as the conversion kinetics between the polysulfides [115].

Based on a similar concept, activated cotton textile was adopted as a scaffold to load Fe/Fe_3_C-encapsulated multiwalled carbon nanotubes via a strategy combining vapor–liquid–solid and solid–liquid–solid processes (Figure 15a). The as-prepared composite was employed as a free-standing interlayer for LIBs, which leads to high cycling stability, ultralow capacity decay rate and remarkable specific capacity due to the enhanced electrode stability and suppressed shuttle effect of polysulfides [117]. In addition, phytoremediation residue-derived carbons were used as a host for LSBs. After the phosphorous acid-assisted pyrolysis of oilseed rape stems from phytoremediation, the as-yield carbon materials with a porous structure and abundant N, P, O doping sites could effectively enhance the sulfur loading amount and polysulfide adsorption, which lead to excellent electrochemical performances. This research not only proposed a promising approach for the safe disposal of phytoremediation residues but also high-performance cathode materials for LSBs [118]. As depicted in Figure 15b, carbon materials were prepared from brewing waste without any activation process. These carbons demonstrate high surface area and interconnected micro- and mesoporous distributions, which raises the capacity values and cyclability of LSBs. This work demonstrated a promising and sustainable way to yield porous carbons while adopting a simple process without activation process.

## 5. Conclusions and Outlook

Due to their low cost, abundance in resources and environmentally friendly qualities, biomasses generated from different scenarios have been employed to prepare carbon matrices as host or interlayer materials for LSBs. Many case works have proven that a hierarchical porous structure and heteroatom-doping sites inherited from the tissue of the biomass used are the main advantages to enhance sulfur utilization, prohibit the shuttle effect of polysulfides, and accelerate the sluggish redox conversion of soluble intermediates. Although remarkable research works have been devoted to the rational design and preparation of biomass-derived carbons, several bottlenecks still remain and require further investigation. The first challenge is that the activation of carbons usually needs corrosive or even poisonous chemical regents, such as NaOH, KOH, and ZnCl_2_, which hinders the commercialization of biomass derived carbons. Although previous works have been devoted to preparing biomass-derived carbons without further activation or purification [119,120], more efforts should be devoted to developing green and low-toxic pore regulation strategies. Secondly, many of the reported sources of biomasses are rather limited in quantity or unstable in quality, which usually exhibit unexpected variation in their structure and composition, resulting in adverse impacts on the standard production of such carbon materials. The third obstacle is the inadequate test equipment used to characterize the structure evolution of biomass-derived carbons during their charge and discharge processes, which makes the underlying work mechanism of these carbons in LSBs difficult to be revealed. The final problem is the relative low loading amount of active species on the cathode of LSBs. It has been reported that the S loading amount could be enhanced to above 2 mg cm^−2^ [121,122] or even 5 mg cm^−2^ [123,124,125]. However, there is still a distance to meet the requirement for real application scenarios.

Based on these aforementioned challenges, future efforts in the design and production of biomass-derived carbons should be focused on the following aspects. First of all, low-cost, low-energy-consumption and ecofriendly strategies should be further explored for the massive production of biomass-derived carbons. For example, mild activation methods should be developed to reduce hazardous emission, potential safety concerns and corrosion on the equipment used. In addition, more fundamental understanding should be realized regarding the activation processes, such as the doping process of heteroatoms, the surface modification of different functional groups, and the regulation of porous structures and their conductivities. Further exploration of the formation mechanism of biomass-derived carbons could provide more specific instruction on the potential mass production of these high-performance carbons.

Secondly, in order to achieve the commercialization of biomass-derived carbons for LSBs, a standard production process should be established based on biomasses with high yield and abundant resources. For example, the composition and structure of biomasses vary dramatically as they are harvested in different seasons or at different locations. Even with the same biomass, different preparation methods usually lead to different structural qualities of carbons. Therefore, in order to provide stable and high-performance carbons for LSBs, many standards should not only be established for the collection and pretreatment of biomasses but also for the pyrolysis process and posttreatment.

Thirdly, advanced equipment composed of microscopy, spectroscopy and X-ray absorption should be designed and assembled to observe in situ the structure, composition and morphology evolution of biomass-derived carbons during the operation of LSBs, which could help to accurately select the appropriate carbons to attain a high-performance cathode or interlayer for LSBs. In particular, the characterization of the electrolyte’s decomposition and conversion on the surface of the carbon electrode could provide straightforward evidence to enhance the coulombic efficiency, rate performance and cycling stability of LSBs.

Finally, the production of carbons from biomasses not only reduce the environmental burden from biomasses or biowastes generated in different industries but also pave an ecofriendly way for backing up the utilization of renewable energy with high-performance energy storage devices. However, an unneglectable fact that should not be omitted is that the activation, pyrolysis and purification processes during the production generate a large volume of liquid and gaseous effluents, and require high energy and chemical reagent consumptions, which will again lead to a huge environmental burden. Therefore, systematic environmental and economic sustainability studies should be devoted to the green and standardized mass production of biomass-derived carbons, which could provide more opportunities for the final commercialization of biomass carbon-derived LSBs.

## Figures and Tables

**Figure 1 nanomaterials-13-01768-f001:**
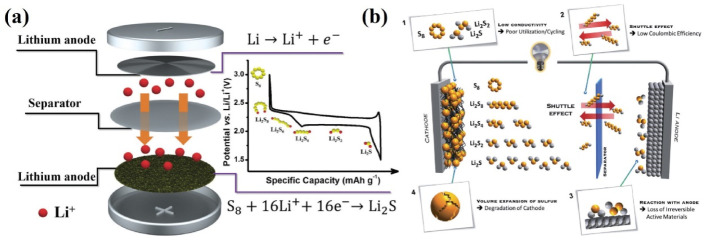
(**a**) Structure of an LSB and the typical charge–discharge process and intermediate product conversion of the battery [53]. Reprinted with permission from Ref. [53]. Copyright 2018 Wiley-VCH. (**b**) Conversion of soluble polysulfide and insoluble Li_2_S_2_/Li_2_S during charge and discharge in a secondary lithium–sulfur battery consisting of a sulfur positive electrode and lithium negative electrode [8]. Reprinted with permission from Ref. [8]. Copyright 2017 Wiley-VCH.

**Figure 2 nanomaterials-13-01768-f002:**
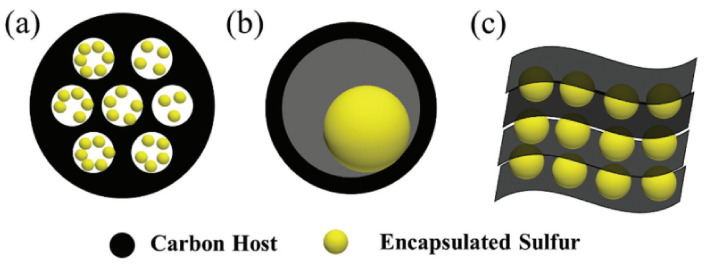
Carbon-based sulfur carriers with different nanostructures: (**a**) porous structure; (**b**) hollow structure.; and (**c**) lamellar graphene structure [53]. Reprinted with permission from Ref. [53]. Copyright 2018 Wiley-VCH.

**Figure 3 nanomaterials-13-01768-f003:**
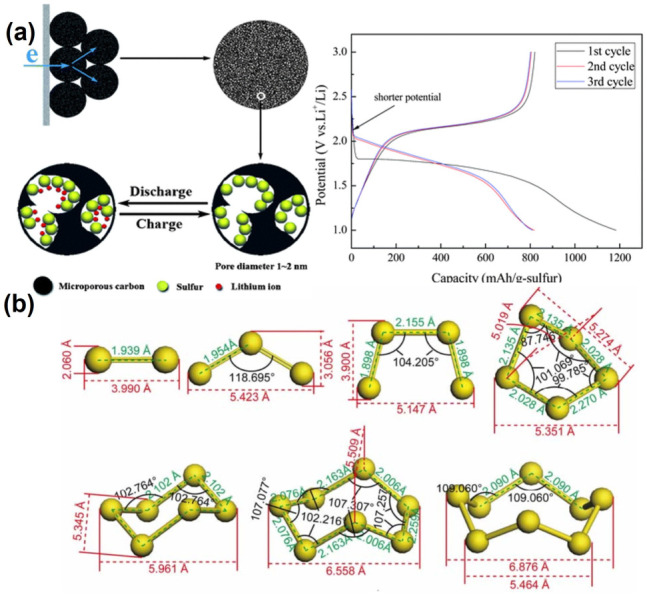
(**a**) The structure diagram and the discharge curve of microporous carbon spheres [56]. Reprinted with permission from Ref. [56]. Copyright 2008 Royal Society of Chemistry. (**b**) Models of various polysulfides formed during the charge and discharge process [7]. Reprinted with permission from Ref. [7]. Copyright 2012 American Chemical Society.

**Figure 4 nanomaterials-13-01768-f004:**
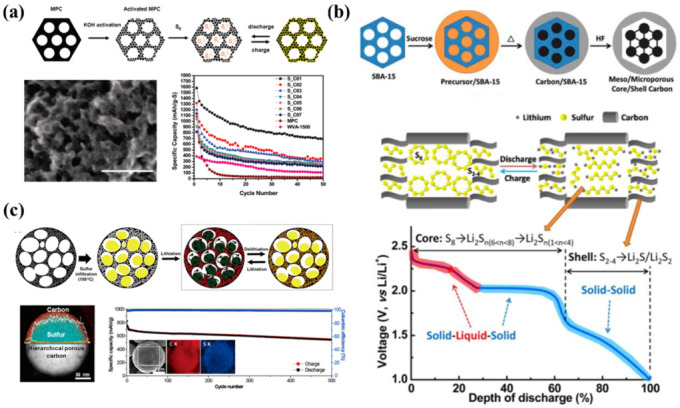
(**a**) The synthesis route, SEM images and corresponding cycle performance curves of carbon materials with different pore size distributions [58]. Reprinted with permission from Ref. [58]. Copyright 2009 American Chemical Society (the scale bar represents 50 nm). (**b**) Structure of core–shell hierarchical porous carbon materials and corresponding electrode reactions of different discharge platforms [60]. Reprinted with permission from Ref. [60]. Copyright 2009 American Chemical Society. (**c**) Structure of hierarchical porous carbon to obtain high sulfur accommodation and the cyclic performance curve at high current density [59]. Reprinted with permission from Ref. [59]. Copyright 2014 American Chemical Society.

**Figure 5 nanomaterials-13-01768-f005:**
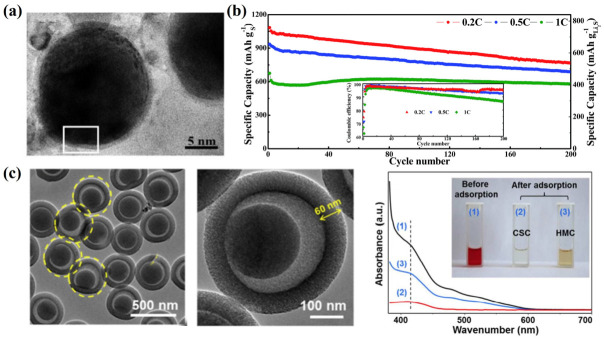
(**a**) TEM images of the core–shell structure Li_2_S@C nanocomposites [61]. Reprinted with permission from Ref. [61]. Copyright 2012 Royal Society of Chemistry (the white square highlights the carbon layer on the Li_2_S particle). (**b**) The cyclic performances of the nanocore–shell Li_2_S@C composites [62]. Reprinted with permission from Ref. [62]. Copyright 2012 Royal Society of Chemistry. (**c**) TEM images and anchoring ability for polysulfides of hollow yolk-and-shell carbon spheres [63]. Reprinted with permission from Ref. [63]. Copyright 2017 American Chemical Society.

**Figure 6 nanomaterials-13-01768-f006:**
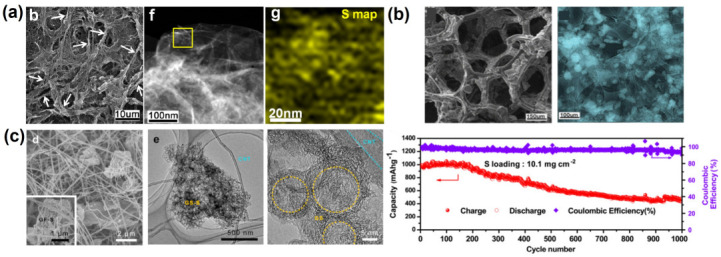
(**a**) SEM, TEM images and the corresponding S element map of the fibrous graphene/sulfur composites [64]. Reprinted with permission from Ref. [64] (the white arrows represent the fibrous graphene, and the yellow square indicates the sulfur map region of the samples). Copyright 2013 American Chemical Society. (**b**) The SEM and element map images of the self-supported graphene foam electrode (top), as well as their cycling performances of the corresponding LSB with high S loading density. [65]. Reprinted with permission from Ref. [65]. Copyright 2014 Elsevier Ltd. (**c**) SEM and TEM images of the carbon nanotube-supported hollow graphene spheres [66]. Reprinted with permission from Ref. [66]. Copyright 2014 Elsevier Ltd (the yellow circles represent the hollow graphene spheres).

**Figure 7 nanomaterials-13-01768-f007:**
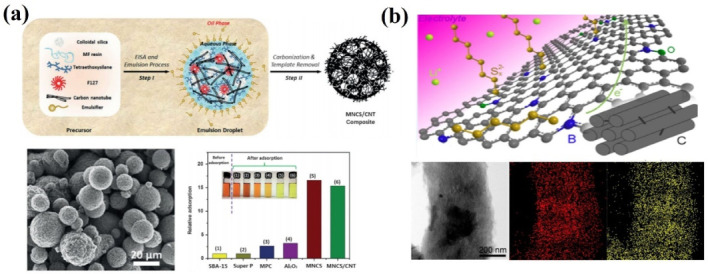
(**a**) Preparation routes, SEM images and polysulfide anchoring ability of carbon nanotube-penetrated nitrogen-doped mesoporous carbon microspheres [67]. Reprinted with permission from Ref. [67]. Copyright 2015 Wiley-VCH. (**b**) Mechanism diagram, STEM image and element distribution diagram of boron-doped ordered porous carbon/sulfur electrode material [68]. Reprinted with permission from Ref. [68]. Copyright 2014 American Chemical Society.

**Figure 8 nanomaterials-13-01768-f008:**
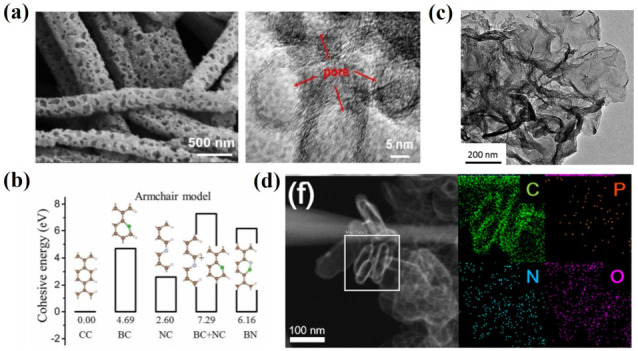
(**a**) SEM and TEM images of nitrogen–oxygen co-doped porous carbons [69]. Reprinted with permission from Ref. [69]. Copyright 2018 American Chemical Society. (**b**), The binding energy of polysulfide with different element doping patterns of the N-B co-doped curved graphene nanoribbon [70]. Reprinted with permission from Ref. [70]. Copyright 2012 Royal Society of Chemistry. (**c**) TEM image of nitrogen–sulfur co-doped porous graphene [71]. Reprinted with permission from Ref. [71]. Copyright 2012 Royal Society of Chemistry. (**d**) TEM image and element distribution of nitrogen, oxygen and phosphorus co-doped hollow carbon nanocapsules/graphene composites [72]. Reprinted with permission from Ref. [72]. Copyright 2018 American Chemical Society.

**Figure 9 nanomaterials-13-01768-f009:**
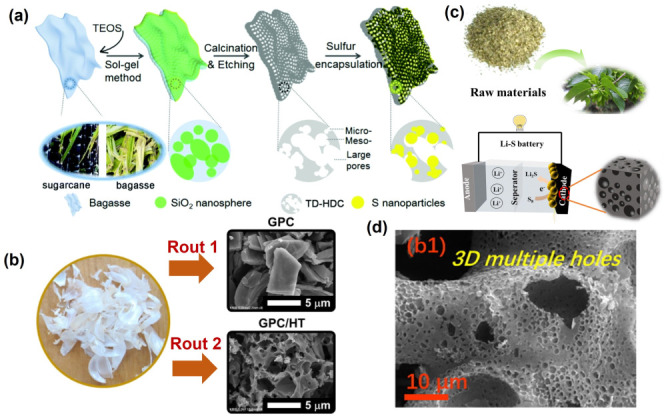
(**a**) Schematic illustration of the formation process of bagasse-derived carbon/sulfur composite [75]. Reprinted with permission from Ref. [75]. Copyright 2021 Royal Society of Chemistry. (**b**) Preparation of porous carbons using garlic peels and the SEM images of the carbons derived from different preparation routs [79]. Reprinted with permission from Ref. [79]. Copyright 2021 Elsevier Ltd. (**c**) The schematic illustration for the synthesis of Eucommia leaf residue-derived hierarchical porous carbon and their application in LSBs [82]. Reprinted with permission from Ref. [82]. Copyright 2023 Elsevier Ltd. (**d**) The SEM image of oxygen-doped carbon derived from balsa waste [83]. Reprinted with permission from Ref. [83]. Copyright 2021 Elsevier Ltd.

**Figure 10 nanomaterials-13-01768-f010:**
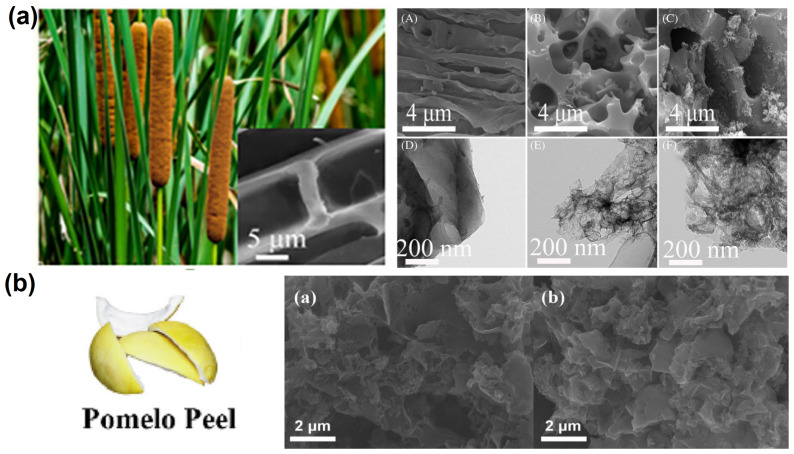
(**a**) Photograph of the cattail biomass, SEM and TEM images of the cattail derived carbons prepared based on different processes. [84] Reprinted with permission from Ref. [84]. Copyright 2022 John Wiley and Sons. (**b**) Illustration of the pomelo peels, and the SEM images of the as-synthesized carbons before (left) and after (right) sulfur loading [88]. Reprinted with permission from Ref. [88]. Copyright 2020 Elsevier Ltd.

**Figure 11 nanomaterials-13-01768-f011:**
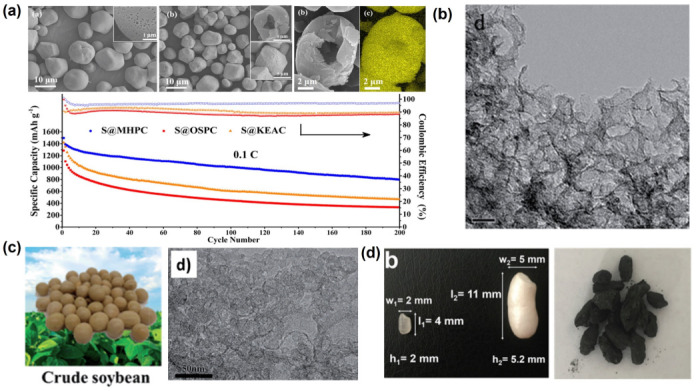
(**a**) Morphology of the hollow microporous carbon spheres prepared from starch via a multi-step pyrolysis method and their cycling performance at 0.1 C [97]. Reprinted with permission from Ref. [97]. Copyright 2017 American Chemical Society. (**b**) The TEM image of nitrogen–sulfur co-doped carbon materials prepared from soluble starch via a template method [98]. Reprinted with permission from Ref. [98]. Copyright 2012 Royal Society of Chemistry (The scale bar represents 20 nm). (**c**) Photograph of the crude soybean biomass and the TEM image of the nitrogen–oxygen co-doped porous carbon materials prepared by using soybean as a precursor [100]. Reprinted with permission from Ref. [100]. Copyright 2016 Royal Society of Chemistry. (**d**) The photograph of the rice before and after expansion treatment, as well as the photograph of the nickel-doped hybrid carbon prepared based on the expansion technology and calcination [99]. Reprinted with permission from Ref. [99]. Copyright 2017 WILEY-VCH.

**Figure 12 nanomaterials-13-01768-f012:**
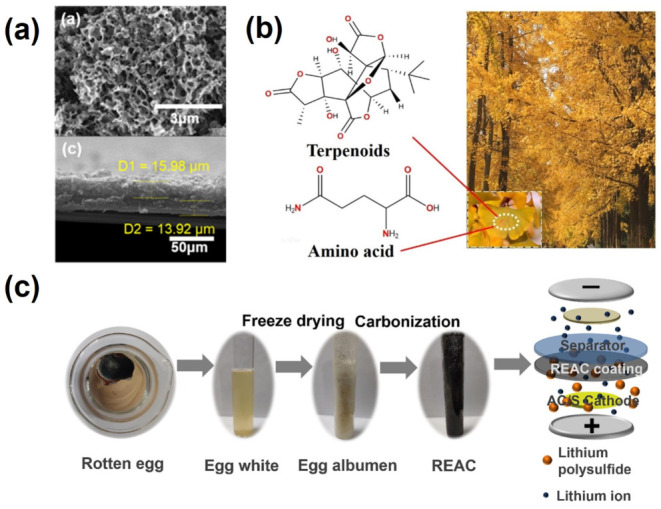
(**a**) SEM images of the top and cross-section morphology of chlorella-derived carbon-coated separator [101]. Reprinted with permission from Ref. [101]. Copyright 2020 Elsevier Inc. (**b**) Chemicals contained in the Ginkgo Folium biomass [103]. Reprinted with permission from Ref. [103]. Copyright 2022 Elsevier Inc. (**c**) Synthesis process of rotten egg-derived carbon-modified separator [104]. Reprinted with permission from Ref. [104]. Copyright 2021 Elsevier Inc.

**Figure 13 nanomaterials-13-01768-f013:**
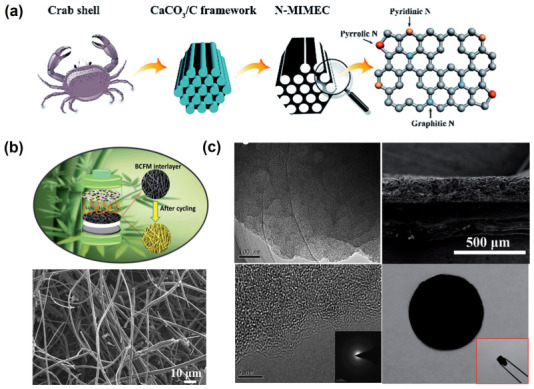
(**a**) Preparation route and the N doping sites of the hierarchical porous carbon materials prepared from crab shells [105]. Reprinted with permission from Ref. [105]. Copyright 2018 Royal Society of Chemistry. (**b**) Fabrication scheme and SEM image of hollow carbon fiber networks prepared from discarded bamboo chopsticks [106]. Reprinted with permission from Ref. [106]. Copyright 2012 Royal Society of Chemistry. (**c**) TEM, SEM images and the photograph of the self-supporting sulfur-doped porous carbon sandwich membrane prepared from loofah pulp [107]. Reprinted with permission from Ref. [107]. Copyright 2012 Royal Society of Chemistry.

**Figure 14 nanomaterials-13-01768-f014:**
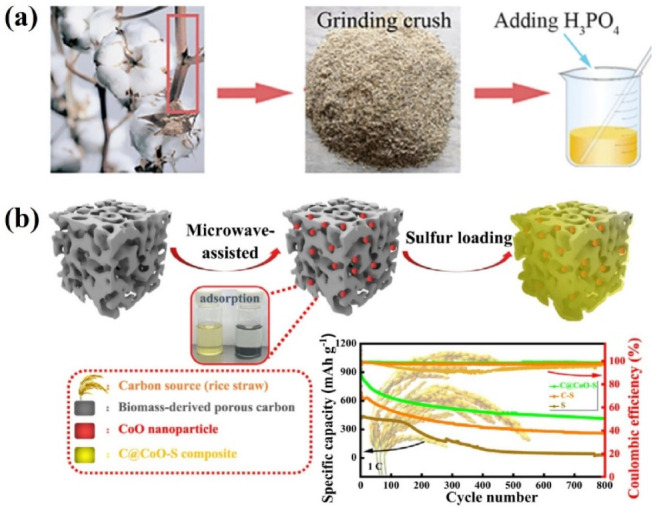
(**a**) Schematic illustration of the preparation procedure of P-doped carbon materials from cotton stalk biowaste [108]. Reprinted with permission from Ref. [108]. Copyright 2022 American Chemical Society. (**b**) Highly dispersed CoO nanoparticles are formed in the inner channels of biochar via microwave-assisted treatment. [115] Reprinted with permission from Ref. [115]. Copyright 2023 Elsevier Inc.

**Figure 15 nanomaterials-13-01768-f015:**
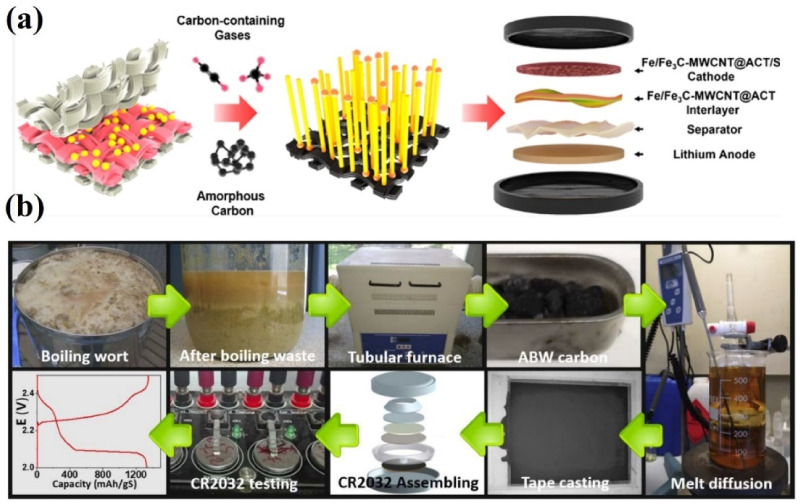
(**a**) Schematic of cotton-derived Fe/Fe_3_C-encapsulated multiwalled carbon nanotubes’ interlayer and their application in LSBs [117]. Reprinted with permission from Ref. [117]. Copyright 2022 American Chemical Society. (**b**) Scheme of cathode fabrication from beer production waste for LSBs [119]. Reprinted with permission from Ref. [119]. Copyright 2020 Wiley-VCH.

## Data Availability

The data is available upon request.

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
