# Peer review of "Green Production of Biomass-Derived Carbon Materials for High-Performance Lithium–Sulfur Batteries"

_nanomaterials, 2023, doi:10.3390/nano13111768_

Round 1

Reviewer 1 Report

This work deals overviewing of recent advances in using biomass-derived C materials for performance of LSBs.

It stresses the environmental character of biomass and its use as green and renewable source of carbon-based materials for producing LSBs.

The environmental topic of battery production is actual and does deserve attention for the reviewing process.

I value especially the detail figures provided.

REMARKS

1 Take into consideration the upgrade citing the paper dealing with graphene material for the removal purposes. Follow, the down below upgrade

“Duo to their high electroconductivity, thermal and chemical stability as well as the bio-compatibility, many types of carbon materials such as graphene[4-9, https://doi.org/10.1007/s11164-018-3698-0], carbon fi-bers[10-12], carbon dots[13], active carbons[14-16] as well as biomass derived carbons[17-26] were deliberately prepared for different applicational territories.“

2 Make a brief pros and cons table to confront the different carbon materials and comparing their advantages vs. disadvantages in battery production.

3 Provide the feasible future aims of the authors as for this scope of investigation.

Author Response

Reviewer 1

This work deals overviewing of recent advances in using biomass-derived C materials for performance of LSBs. It stresses the environmental character of biomass and its use as green and renewable source of carbon-based materials for producing LSBs.

The environmental topic of battery production is actual and does deserve attention for the reviewing process. I value especially the detail figures provided.

REMARKS

1 Take into consideration the upgrade citing the paper dealing with graphene material for the removal purposes. Follow, the down below upgrade

“Duo to their high electroconductivity, thermal and chemical stability as well as the bio-compatibility, many types of carbon materials such as graphene[4-9, https://doi.org/10.1007/s11164-018-3698-0], carbon fi-bers[10-12], carbon dots[13], active carbons[14-16] as well as biomass derived carbons[17-26] were deliberately prepared for different applicational territories.“

Reply: Thanks for your kind reminding, and we have revised this section based on your suggestion and a new citation was made based on your suggestion. Please find them in the revised version.

2 Make a brief pros and cons table to confront the different carbon materials and comparing their advantages vs. disadvantages in battery production.

Reply: Thanks for your comments, we have added two tables to summarize the pros and cons of the materials in terms of preparation method.

Please find the tables in the revised manuscript.

3 Provide the feasible future aims of the authors as for this scope of investigation.

Reply: Thanks for your comments, we have added future aims for the application of biomass derived carbon for LSBs in the prospective section.

Reviewer 2 Report

·        * In the introduction, the authors mention the search for 'advanced energy systems'. references 1 and 2 concern only one system. The fact is that many energy systems are being studied. I suggest the authors add one or more references that also concern the other "advanced energy system".

·         *The theoretical capacity of Li-S batteries is 1675 mAhg-1 and not 1672 mAhg-1 as mentioned in the last line of the first page.

·        * Ref. [4,7,9] regards Graphene oxide materials.

·         *Ref[36] regards Lithium-Ion batteries and not Li-S batteries. I suggest the authors move it to Lithium-Ion batteries references (with Ref. [34]).

·         *Figure 1(a): I suggest that the authors add the abscissa axis to Figure 1(a).

·        * Figure 3: the resolution of the figure is too low. I suggest the authors improve it.

·       *  Figure 5: the resolution of the figure is too low. I suggest the authors improve it.

·         *Figure 6: the resolution of the figure is too low. I suggest the authors improve it.

·         *Figure 7: the resolution of the figure is too low. I suggest the authors improve it.

·         *Figure 8: the resolution of the figure is too low. I suggest the authors improve it.

·       *  Figure 13: the resolution of the figure is too low. I suggest the authors improve it.

·        * The article needs more comprehensive comparison tables with details of the performances of materials treated in the paper.

·         *The language of the paper needs to be improved.

The language of the paper needs to be improved.

Author Response

Reviewer 2

* In the introduction, the authors mention the search for 'advanced energy systems'. references 1 and 2 concern only one system. The fact is that many energy systems are being studied. I suggest the authors add one or more references that also concern the other "advanced energy system".

Reply: Based on your suggestion, we have added more references concerning about the other energy systems in this section.

*The theoretical capacity of Li-S batteries is 1675 mAhg-1 and not 1672 mAhg-1 as mentioned in the last line of the first page.

Reply: Sorry for this mistake, and we have corrected them up in the revised version.

* Ref. [4,7,9] regards Graphene oxide materials.

Reply: Thanks for your comments, we have revised these citations with other ones concerning about the biomass carbons.

*Ref[36] regards Lithium-Ion batteries and not Li-S batteries. I suggest the authors move it to Lithium-Ion batteries references (with Ref. [34]).

Reply: Thanks for your careful checking, we have changed the sequent of the references based on your suggestion.

*Figure 1(a): I suggest that the authors add the abscissa axis to Figure 1(a).

  Figure 3: the resolution of the figure is too low. I suggest the authors improve it.

Figure 5: the resolution of the figure is too low. I suggest the authors improve it.

Figure 6: the resolution of the figure is too low. I suggest the authors improve it.

Figure 7: the resolution of the figure is too low. I suggest the authors improve it.

Figure 8: the resolution of the figure is too low. I suggest the authors improve it.

Figure 13: the resolution of the figure is too low. I suggest the authors improve it.

Reply: thanks for your comment on these figures, we have revised them according to your suggestion, please find them in the revised manuscript.

For Figure 1, the figure is quoted from the literature, we cannot add additional information in this figure. Moreover, we have also provided new images with high quality to replace this image.

Thanks for your kind checking on figure 3,5,6,7,8 and 13, based on your suggestion, we have carefully revised these figures in the revised version. Based on these modifications on these images, the figure captions were also changed.

The article needs more comprehensive comparison tables with details of the performances of materials treated in the paper.

Reply: Thanks for your suggestion, we have added two comparison tables in the revised version to highlight the difference of the biomass derived materials.

Please find the detailed comparison of the biomass derived materials in table 1 and table 2 in the revised manuscript.

Please find the tables in the revised manuscript.

*The language of the paper needs to be improved.

Reply: Thanks for your comments, we have revised the language of the manuscript with the help of several friend who have been working oversea for many years and publish many research works in high reputation journals. Based on their help, many ill-structured sentences, grammar mistakes as well as spelling errors were corrected to make the language of the manuscript much enhanced.

All the part received correction were marked in red color.

Reviewer 3 Report

This work reports a review on the use of carbons derived from biomass for cathodes and interlayer in Lithium-Sulfur batteries (LSB). In the first section, the authors put into context the need for carbons to alleviate the intrinsic problems of this battery technology during charge/discharge operation. In this section, a review of different types of carbons is briefly addressed. In the main section of the manuscript, recent advances on different biomass-derived carbons integrated into LSB are exposed and graphically summarized. The authors close the article with some final considerations on the pending challenges of this topic. The manuscript is correctly stated and may be of interest as a review study for researchers in the same field. However, before its publication, the authors should take into account the following recommendations in order to increase the quality of the manuscript:

(1) In section 4.3, the authors could delve into one of the critical aspects in the synthesis of biomass-derived carbons for LSS: the activation process. For this, the authors could add a comparative table collecting the results of works that use different activation processes. This table could describe the synthesis conditions, the textural properties of the coals, and the electrochemical performance in LSB (with the electrode properties, as %S).

(2) In section 4.4, authors should add and comment on additional examples of biomass-derived carbons modified with different compounds such as NiO [10.1007/s11581-021-04275-8], TiO2 [10.1016/j.jpowsour.2016.02.061], Ti4O7 [10.1038/ncomms5759], MnO2 [10.1016/j.electacta.2018.09.176], CoO [10.1016/j.jcis.2023.02.123] or even with conductive polymers [10.1039/d1se02052h].

(3) In section 5 (conclusion and Outlook), the authors rightly state the following: “the future efforts in the design and production of biomass derived carbons should be focused on the following aspects. First of all, low-cost, low energy consumption and eco-friend strategies should be further explored for the massive production of the coals. For example, mild activation methods should be developed to reduce the hazardous emission, the potential safety concerns as well as the corrosion on the equipment.” This path of study has already been initiated, detecting a first work where the preparation of a carbon derived from biomass (agri-food industry) without activation or purification stage is reported [10.1002/cssc.202000969]. Along the same lines, but as a different alternative, activation processes can be avoided if you opt for the use of biomass-derived carbons from other industrial uses, such as filters used for industrial purification of liquids or gases. The success of this innovative solution has already been demonstrated [10.1002/cssc.202101231]. This relevant information should be added and commented on in the text.

(4) In section 5, the authors should add a fourth obstacle: “the low sulfur mass loading of cathodes based on biomass-derived carbons”. In most of the articles on this topic, the mass sulfur loading (mgS/cm2) does not reach the values required for a commercial cell cathode. This challenge is already being addressed by different teams, with > 2mgS/cm2 [10.3390/ma15248856; 10.3390/nano10050840]; or even > 5 mgS/cm2 [10.1007/s11431-022-2226-8; 10.1002/cssc.202202095; 10.1021/acsaem.2c03018]. The discussion about this challenge and the proposed solutions should be discussed in the text.

(5) In section 5, the authors should focus on a problem common to all the articles published on biomass-derived carbons for LSB: the absence of true environmental and economic sustainability studies regarding the process of valorization of said biomasses. Most of these articles use activation, pyrolysis and purification methods, generating a huge volume of liquid and gaseous effluents, as well as high energy consumption and chemical reagents. These problematic aspects are hardly addressed, and yet they are crucial to be able to implement these carbons in a future industrial application in the manufacture of cathodes or interlayers for LSB.

Author Response

Reviewers 3

This work reports a review on the use of carbons derived from biomass for cathodes and interlayer in Lithium-Sulfur batteries (LSB). In the first section, the authors put into context the need for carbons to alleviate the intrinsic problems of this battery technology during charge/discharge operation. In this section, a review of different types of carbons is briefly addressed. In the main section of the manuscript, recent advances on different biomass-derived carbons integrated into LSB are exposed and graphically summarized. The authors close the article with some final considerations on the pending challenges of this topic. The manuscript is correctly stated and may be of interest as a review study for researchers in the same field. However, before its publication, the authors should take into account the following recommendations in order to increase the quality of the manuscript:

(1) In section 4.3, the authors could delve into one of the critical aspects in the synthesis of biomass-derived carbons for LSS: the activation process. For this, the authors could add a comparative table collecting the results of works that use different activation processes. This table could describe the synthesis conditions, the textural properties of the coals, and the electrochemical performance in LSB (with the electrode properties, as %S).

Reply: Thanks a lot for your comments, we have added two comparative tables in this section to highlight the structure variation and electrochemical performance of LSBs.

Please find the tables in the revised manuscript.

(2) In section 4.4, authors should add and comment on additional examples of biomass-derived carbons modified with different compounds such as NiO [10.1007/s11581-021-04275-8], TiO2 [10.1016/j.jpowsour.2016.02.061], Ti4O7 [10.1038/ncomms5759], MnO2 [10.1016/j.electacta.2018.09.176], CoO [10.1016/j.jcis.2023.02.123] or even with conductive polymers [10.1039/d1se02052h].

Reply: thanks for your suggestion, we have added the aforementioned citations in the revised version of the manuscript in section 4.4.

[111]Zhu, M. L.; Wu, J.; Li, S. Q.; Flower‑like Ni/NiO microspheres decorated by sericin‑derived carbon for high‑rate lithium‑sulfur batteries .Ionics. 2021, 27 : 5137–5145. https://doi.org/10.1007/s11581-021-04275-8

[112]Moreno, N.; Caballero, Á.; Morales, J.; Improved performance of electrodes based on carbonized olive stones/S composites by impregnating with mesoporous TiO2 for advanced Li-S batteries. J Power Sources. 313, 2016, 21-29. http://dx.doi.org/10.1016/j.jpowsour.2016.02.061

[113]Pang, Q.; Kundu, D.; Cuisinier, M.; Surface-enhanced redox chemistry of polysulphides on a metallic and polar host for lithium-sulphur batteries. Nat Commun. 2014, 5:4759. http://www.nature.com/naturecommunications

[114]Luna-Lama, F.; Hernández-Rentero, C.; Caballero, A.; Biomass-derived carbon/g-MnO2 nanorods/S composites prepared by facile procedures with improved performance for Li/S batteries. Electrochim Acta. 292, 2018, 522-531. https://doi.org/10.1016/j.electacta.2018.09.176

[115]Wang, J.; Wu, L.; Shen, L.; CoO embedded porous biomass-derived carbon as dual-functional host material for lithium-sulfur batteries. J Colloid Interf Sci. 640, 2023, 415–422. https://doi.org/10.1016/j.jcis.2023.02.123

[116]Lama, F. L.; Caballero, Á.; Morales, J.; Synergistic effect between PPy:PSS copolymers and biomass-derived activated carbons: a simple strategy for designing sustainable highperformance Li–S batteries. Sustain Energy Fuels. 2022, 6, 1568–1586. https://doi.org/10.1039/rsc_crossmark_policy

(3) In section 5 (conclusion and Outlook), the authors rightly state the following: “the future efforts in the design and production of biomass derived carbons should be focused on the following aspects. First of all, low-cost, low energy consumption and eco-friend strategies should be further explored for the massive production of the coals. For example, mild activation methods should be developed to reduce the hazardous emission, the potential safety concerns as well as the corrosion on the equipment.” This path of study has already been initiated, detecting a first work where the preparation of a carbon derived from biomass (agri-food industry) without activation or purification stage is reported [10.1002/cssc.202000969]106.1. Along the same lines, but as a different alternative, activation processes can be avoided if you opt for the use of biomass-derived carbons from other industrial uses, such as filters used for industrial purification of liquids or gases. The success of this innovative solution has already been demonstrated [10.1002/cssc.202101231]106.2. This relevant information should be added and commented on in the text.

Although serval case works have been devoted to yield biochar without activation process or with mild activation process, inducst

Reply: Thanks for your kind reminding, we have revised this section and the two references were added to backup our statements in the outlook section.

[119] Tesio, A. Y.; Gómez-Camer, J. L.; Morales, J.; Simple and sustainable preparation of non-activated porous carbon from brewing waste for high-performance lithium–sulfur batteries. ChemSusChem. 10.1002/cssc.202000969. https://doi.org/10.1002/cssc.202000969

[120] Benítez, A.; Márquez, P.; M. Martín, Á.; Simple and Sustainable Preparation of Cathodes for Li–S Batteries: Regeneration of Granular Activated Carbon from the Odor Control System of a Wastewater Treatment Plant. ChemSusChem. 2021, 14, 3915–3925. https://doi.org/10.1002/cssc.202101231

(4) In section 5, the authors should add a fourth obstacle: “the low sulfur mass loading of cathodes based on biomass-derived carbons”. In most of the articles on this topic, the mass sulfur loading (mgS/cm2) does not reach the values required for a commercial cell cathode. This challenge is already being addressed by different teams, with > 2mgS/cm2 [10.3390/ma15248856; 10.3390/nano10050840]106.3-106.4; or even > 5 mgS/cm2 [10.1007/s11431-022-2226-8; 10.1002/cssc.202202095; 10.1021/acsaem.2c03018].106.5-106.7 The discussion about this challenge and the proposed solutions should be discussed in the text.

Reply: Many thanks for your critical comment, this point is the bottle neck in the LSBs system. Thanks a lot and we have added these in the revised version. And all the mentioned references were cited in the revised manuscript to back up our statement.

[121] Páez Jerez, A. L.; Mori, M. F.; Flexer, V.; Water Kefir Grains—Microbial Biomass Source for Carbonaceous Materials Used as Sulfur-Host Cathode in Li-S Batteries. Materials. 2022, 15, 8856. https://doi.org/10.3390/ma15248856

[122] Benítez, A.; Morales, J.; Caballero, Á.; Pistachio Shell-Derived Carbon Activated with Phosphoric Acid: A More Efficient Procedure to Improve the Performance of Li–S Batteries. Nanomaterials. 2020, 10, 840. https://doi.org/10.3390/nano10050840

[123] Liu, L. Z.; Xia, G. H.; Wang, D.; Biomass-derived self-supporting sulfur host with NiS/C composite for high-loading Li-S battery cathode. Sci China Tech Sci. 2023, 66: 181–192. https://doi.org/10.1007/s11431-022-2226-8

[124] Lama, F. L.; Marangon, V.; Caballero, Á.; Diffusional Features of a Lithium-Sulfur Battery Exploiting Highly Microporous Activated Carbon. ChemSusChem. 2023, 16, e202202095 (1 of 18). https://doi.org/10.1002/cssc.202202095

[125] Liu, L. Z.; Xia, G. H.; Wang, D.; Self-supporting Biomass Li−S Cathodes Decorated with Metal Phosphides−Higher Sulfur Loading, Better Stability, and Longer Cycle Life. ACS Appl Energy Mater. 2022, 5, 15401−15411. https://doi.org/10.1021/acsaem.2c03018.

(5) In section 5, the authors should focus on a problem common to all the articles published on biomass-derived carbons for LSB: the absence of true environmental and economic sustainability studies regarding the process of valorization of said biomasses. Most of these articles use activation, pyrolysis and purification methods, generating a huge volume of liquid and gaseous effluents, as well as high energy consumption and chemical reagents. These problematic aspects are hardly addressed, and yet they are crucial to be able to implement these carbons in a future industrial application in the manufacture of cathodes or interlayers for LSB.

Reply: many thanks for your critical comment, this point is the Achilles' Heel in the synthesis of the biomass carbons. Thanks a lot, and we have reminded this critical point in the final section of the revised manuscript.

Round 2

Reviewer 1 Report

Authors have reacted to the given queries.

Reviewer 2 Report

I would like to thank the authors for revising the paper.

Reviewer 3 Report

The authors have made the required modifications in order to increase the quality of the manuscript.